# Dielectric Metasurface-Based High-Efficiency Mid-Infrared Optical Filter

**DOI:** 10.3390/nano8110938

**Published:** 2018-11-14

**Authors:** Fei Shen, Qianlong Kang, Jingjing Wang, Kai Guo, Qingfeng Zhou, Zhongyi Guo

**Affiliations:** School of Computer and Information, Hefei University of Technology, Hefei 230009, China; shenfei@hfut.edu.cn (F.S.); m18955890051@163.com (Q.K.); wangjingjing341225@163.com (J.W.); enqfzhou@hfut.edu.cn (Q.Z.)

**Keywords:** optical filter, dielectric metasurfaces, mid-infrared (mid-IR)

## Abstract

Dielectric nanoresonantors may generate both electric and magnetic Mie resonances with low optical loss, thereby offering highly efficient paths for obtaining integrated optical devices. In this paper, we propose and design an optical filter with a high working efficiency in the mid-infrared (mid-IR) range, based on an all-dielectric metasurface composed of silicon (Si) nanodisk arrays. We numerically demonstrate that, by increasing the diameter of the Si nanodisk, the range of the proposed reflective optical filter could effectively cover a wide range of operation wavelengths, from 3.8 μm to 4.7 μm, with the reflection efficiencies reaching to almost 100%. The electromagnetic eigen-mode decomposition of the silicon nanodisk shows that the proposed optical filter is based on the excitation of the electric dipole resonance. In addition, we demonstrate that the proposed filter has other important advantages of polarization-independence and incident-angle independence, ranging from 0° to 20° at the resonance dip, which can be used in a broad range of applications, such as sensing, imaging, and energy harvesting.

## 1. Introduction

The mid-infrared (mid-IR) light source is of particular interest because the transmission windows of the Earth’s atmosphere are partially located in the range 3–5 μm (mid-IR) and 8–13 μm (long-IR), thereby offering new possibilities for applications related to military, medical, communication, and solar energy [1,2,3,4]. To take advantage of the mid-IR source, we need to develop high-performance mid-IR elements such as lenses, polarizers, and filters. Conventionally, a mid-IR filter is designed in terms of Fresnel formulas and consists of bulky multilayered dielectric films, hindering their application in compact optical devices. With the development of nanofabrication techniques, a new kind of material (i.e., metasurfaces) is a promising solution, with significant properties of artificially engineered optical response and high integration [5,6,7]. The first proposed metasurface is composed of subwavelength plasmonic nanoantennas, arranged in two-dimensional arrays; therefore, both the amplitude and phase of the incident beam could be artificially controlled through the local resonance of each of the nanoantennas [8,9,10]. However, the intrinsic loss of plasmonic structure in the optical range still hinders the use of metasurfaces as practical filters, even though tremendous efforts for enhancing the efficiency have been made [11,12,13]. On this issue, a high-index all-dielectric metasurface can be an alternative, over plasmonic nanostructures, for its relatively low loss, therefore possessing high-quality multipole resonances [14,15,16] and resulting in a high working efficiency [17,18,19,20]. The relative spectral position of these multipolar eigenmodes can be tailored by careful adjustment of the geometrical parameters. It has been demonstrated that the high-index all-dielectric metasurfaces have the capability of manipulating light-matter interaction with high efficiency, including polarization conversion [21], unidirectional scattering [22], generating vortex beam [23], and molecule sensing [24,25].

In this paper, we propose a high-efficiency dielectric metasurface for a mid-IR filter enabled by the electric dipole resonance. The metasurface consists of a silicon (Si)-disk embedded on a calcium fluoride (CaF_2_) substrate. The optical characterizations show that the all-dielectric mid-IR filter could generate high reflection peaks at designated wavelengths by tuning the diameter of the unit cell. In addition, the reflectance, ranging from 3.8 μm to 4.7 μm, at the electric dipole resonance reaches almost 100%. Furthermore, the filters are independent on both the polarization and incident-angle, and are extremely sensitive to the operating frequency. All results demonstrate the high performance of the proposed dielectric metasurface as a highly efficient and narrow-band mid-IR filter.

## 2. Design, Results, and Discussions

Let us start from the general geometry and operation mechanism of the proposed devices. Figure 1a presents the perspective view of the proposed all-dielectric metasurface, working as a reflective filter. The Si nanodisks were embedded on a calcium fluoride (CaF_2_) substrate, due to their high transmission in the mid-IR range [26]. For this structure, an ultra-smooth CaF_2_ film could be obtained by chemical mechanical polishing [27]. The silicon nanodisk could be fabricated by using an etching process [28]. Compared to other structures, this structure was simple, which is an important advantage for metasurfaces. Figure 1b shows the detailed structure of the unit cell. The lattice constant was set as *p* = 3.6 μm, ensuring that the coupling between isolated Si-disk nanoresonators can be negligible. While varying the nanodisk’s diameter, the height of the resonator was constant at *h* = 0.5 μm. We performed full-wave simulations with a home-built simulation program, based on the finite difference time-domain (FDTD) method, to examine the optical response of the unit cell and the concrete performance of the metasurface as a filter. Periodic boundary conditions were set along the *x* and *y* directions to save simulation time and memory, and perfectly matched layers were defined in the *z* direction for preventing unphysical back-reflected waves from the outer boundaries. In our simulations, the refractive index of Si and CaF_2_ were derived from experimental measurements [26,29]. To accurately calculate results, we set the smallest size of the nonuniform mesh at 10 nm.

Figure 2a shows the dependence of the reflection spectrum on the diameter of the Si nanodisk under normal incidence with *x*-polarization. In general, Si is a highly transmissive material in the mid-IR range. In apparent contrast, the simulated reflection spectrum indicated that the proposed metasurface could support two strong resonances, as shown in Figure 2a. The main peak could cover a broadband wavelength range from 3.8 μm to 4.7 μm, whereas, in contrast, the side peak shifted slowly as the diameter of the Si nanodisk increased. The physical mechanism of the two resonances could be validated by way of comparison between the reflection spectrum and the Cartesian multipole contributions into the scattering cross sections. To this end, the reflection spectrum of the Si nanodisk, with a diameter of D = 1.8 μm, was extracted along the white dashed line in Figure 2a and the plotted line in Figure 2b. In addition, we decomposed the simulated scattering cross section of the Si nanodisk, with a diameter of D = 1.8 μm, into electric and magnetic multipole contributions, as shown in Figure 2c. The Cartesian multipole moments could be calculated from the electric field distributions in the Si disk as per the following [30,31]:
(1)$$p=\varepsilon_{0}(\varepsilon_{d}-\varepsilon_{r})\int E(r)dV$$(2)$$m=\frac{i\omega \varepsilon_{0}(\varepsilon_{d}-\varepsilon_{r})}{2c}\int [r\times E(r)]dV$$(3)$$T=\frac{\omega \varepsilon_{0}(\varepsilon_{d}-\varepsilon_{r})}{10c}\int \left\{\left[r\cdot E(r)]r-2[r\cdot r]E(r\right)\right\}dV$$(4)$$Q_{\alpha \beta }=\frac{\varepsilon_{0}(\varepsilon_{d}-\varepsilon_{r})}{2}\int \left\{r_{\alpha }E_{\beta }+r_{\beta }E_{\alpha }-\frac{2}{3}[r\cdot E(r)]\delta_{\alpha \beta }\right\}dV$$(5)$$M_{\alpha \beta }=\frac{i\omega \varepsilon_{0}(\varepsilon_{d}-\varepsilon_{r})}{3c}\int \{{\left[r\times E(r)\right]}_{\alpha }r_{\beta }+{\left[r\times E(r)\right]}_{\beta }r_{\alpha }\}dV$$

Table 1 summarizes the definitions of the parameters used in Equations (1)–(5). From these near-field distributions, we could calculate the far-field scattering cross section according to the classical electromagnetic field theory [30,32], as shown in Figure 2c. The results in Figure 2b,c show that the reflective spectrum was in good agreement with the Cartesian multipole contributions. It was observed that there was a strong magnetic dipole resonance peak near 3.7 μm and an electric dipole resonance peak at 4 μm. The positions of the electric and magnetic dipole resonance peaks coincided with the position of the reflection peaks in Figure 2b. From the decomposition, we could conclude that the main and side peaks, in the reflective spectrum, were resulted from the electric dipole (λ = 4.3 μm) and magnetic dipole (λ = 3.7 μm), respectively. The reflection at the wavelengths off the electric and magnetic dipoles was close to 0, indicating no light was reflected back. It may also validate that the electric dipole potentially dominates the optical response of the proposed metasurface.

To further elucidate the physical mechanism of the presence of the reflection peaks, we additionally show the electric and magnetic field distributions at wavelengths of 3.7 μm and 4.3 μm in Figure 2d,e, respectively. In Figure 2d, a displacement current loop was present in the *x*–*z* plane and the overall magnetic field strength strongly concentrated at the center of the silicon disk, verifying that the reflection peak at the wavelength of 3.7 μm came from the magnetic dipole resonance. In addition, Figure 2e shows the presence of a strong electric resonance along the *x*-axis, and the electric field energy was mainly located at the center of the silicon cylinder. These appearances were clearly equivalent to an electric dipole mode [33,34].

The above results demonstrated that the proposed dielectric metasurface, enabled by the electric dipole resonance, could be an alternative approach for a mid-IR filter with significant advantages. Because the electric dipole resonance was highly sensitive to the geometrical parameters, a filter for specific wavelengths could be obtained by engineering the diameter of the Si nanodisk. In the following, we turned our attention to the high-efficiency filtering behavior, specifically the dependence of the filtering wavelength on the diameter of the Si nanodisk.

As examples, we selected different Si nanodisk diameters, of 1.2 μm, 1.4 μm, 1.6 μm, 1.8 μm, 2.0 μm, and 2.2 μm, all with the same lattice constant of *p* = 3.6 μm, and studied their behaviors as a mid-IR reflective filter. Figure 3a shows that the filtering wavelength (red) shifted from λ = 3.8 μm to λ = 4.7 μm, with a high reflection efficiency of about 100%. In addition, the full-width at half-maximum (FWHM) of the filtering peak was calculated to be 0.012λ, 0.026λ, 0.043λ, 0.048λ, 0.048λ, and 0.047λ, respective to Si nanodisk diameters, demonstrating a good filtering performance. This was because most of the incident energy at the electric dipole resonance would be reflected, whereas the incident energy off the electric dipole resonance would be filtered out due to the high quality of the electric dipole resonance.

To provide a comprehensive description of the filtering effect, we calculated and plotted the near-field distribution in the *x*–*z* and *x*–*y* planes of the metasurfaces in Figure 3b,c, respectively. The value of E_max_ was the strongest value of electric field at resonance, and other electric field maps were obtain by normalization based on the E_max_. In the numerical simulations, we chose the Si-disk with a diameter of D = 1.8 μm at wavelengths around the reflection peak. It was clear that the Si nanodisk was almost transparent at the wavelengths off the electric dipole resonance, and the electric fields were transmitted through the metasurface, which were the Si characteristics in the mid-IR range. In contrast, the electromagnetic energy at the electric dipole resonance was strongly reflected back from the metasurface, further demonstrating the high-efficiency of the reflective filter.

The optical characterizations show that the designed dielectric filters with different unit sizes generated high reflection peaks at designated wavelengths, with the reflectance mostly reaching to 100% and the reflection peaks containing most of the energy of the whole spectra. Hence, the proposed metasurfaces are able to serve as mid-IR filters.

In order to further investigate the filtering performances, we calculated the dependence of the reflection spectrum on the incident plane wave. Without loss of generality, we took the Si nanodisk with a diameter of D = 1.8 μm as an example. Figure 4a shows that the reflection spectrum remained unchanged when the polarization angle changed, and only two peaks were present, attributed to the electric dipole and magnetic dipole resonances, respectively. The polarization independence was due to the structural symmetry of both the unit cell and lattice. In addition, the proposed metasurface showed excellent angular behavior for the incident angle. To confirm that the metasurface works well when the incident angle increases, the effect of different incident angles on the filtering performance was also investigated. Figure 4b plots the dependence of the reflection spectrum on the incident angle with *x*-polarization. The reflection peak at the electric dipole resonance at the wavelength of 4.3 μm was sustained by the metasurface for the incident angles (*θ*) varying from 0° to 20°, demonstrating that the proposed filter possessed a good tolerance to incident angle variation. Besides the main peak, a new side peak on the reflection spectrum arised at the short wavelength (blue) and shifted when the incident angle increased.

Figure 5a,b demonstrate the near-field distributions of the metasurface at the main peak (~4.3 μm) and side peak (~3.7 μm) in Figure 4b, respectively. As the incident angle increased, the electric field at the wavelength of 4.3 μm was always concentrated at the center of the Si nanodisk, verifying the physical origin of the main peak from the electric dipole resonance. In contrast, the electric field at the wavelength of 3.7 μm was mostly localized at the edge of the Si nanodisk and in the gaps. As known from Figure 2d, it was the magnetic dipole resonance for the normal incidence. As the incident angle increased, the arrangement of the Si nanodisks into a periodic lattice and the oblique wave-vector potentially gave rise to an in-plane propagating collective surface mode, which hybridized with the magnetic dipole resonance. Therefore, the narrow-band mode near the wavelength of 3.7 μm potentially stemmed from the interaction between the magnetic dipole resonance and the collective surface mode, as presented in Figure 5b.

## 3. Conclusions

In conclusion, we proposed a dielectric metasurface for a mid-IR optical filter, operating under the excitation of the electric dipole resonance. The metasurface consists of a Si-disk embedded on a CaF_2_ substrate. The simulation results show that the all-dielectric mid-IR filter could generate high reflection peaks at designated wavelengths by tuning the size of the unit cell. In addition, the maximal reflectance at the electric dipole resonance reached almost 100%. The filters are independent in terms of both the polarization and the incident-angle, due to the symmetry, and are extremely sensitive to the operating frequency. All results demonstrate the high-performance of the proposed dielectric metasurface as a high-efficiency and narrow-band mid-IR filter.

## Figures and Tables

**Figure 1 nanomaterials-08-00938-f001:**
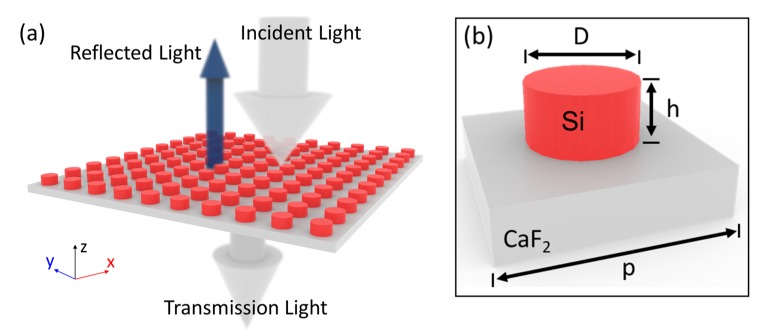
Perspective views of the reflective metasurface as a filter (**a**) and the unit cell (**b**). The lattice constant was *p* = 3.6 μm. The height of the resonator was constant at *h* = 0.5 μm. The diameter of the silicon (Si) nanodisk varied in order to filter different wavelengths.

**Figure 2 nanomaterials-08-00938-f002:**
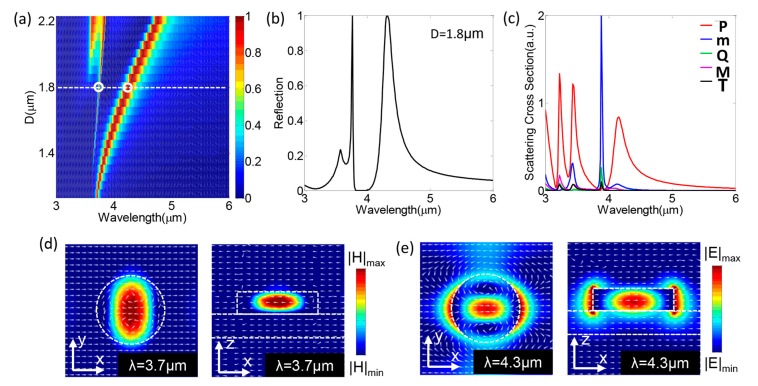
(**a**) The simulated reflection contour image of the filter with different Si-disk diameters from 1.1 μm to 2.2 μm; (**b**) the reflection spectrum extracted along the white dashed line in (**a**); (**c**) the scattering cross section of each of the radiating moments of the examples with D = 1.8 μm; (**d**) the magnetic field in the *x*–*y* plane and the *x*–*z* plane; and (**e**) the electric field in the *x*–*y* plane and the *x*–*z* plane, respectively. The white dashed lines outline the schematic of the nanodisk structure.

**Figure 3 nanomaterials-08-00938-f003:**
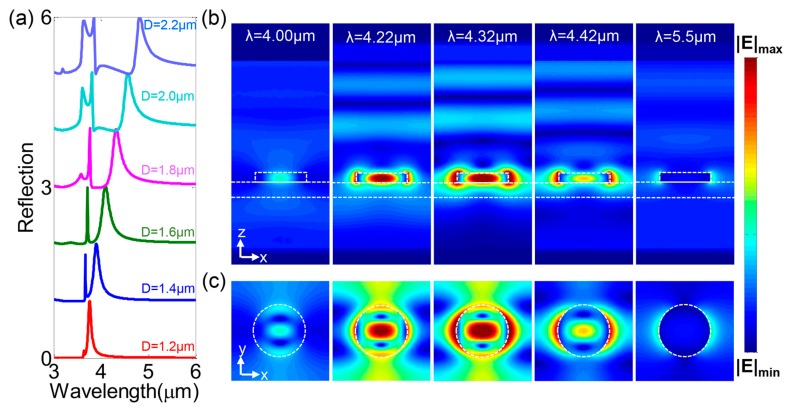
(**a**) The simulated reflection spectra of the proposed filters, where the Si nanodisk diameter ranged from 1.2 μm to 2.2 μm. The electric field distributions in (**b**) *x*–*z* plane and (**c**) *x*–*y* plane for the Si nanodisk with a diameter of D = 1.8 μm around the reflection peak. The white dashed lines outline the schematic of the nanodisk structure.

**Figure 4 nanomaterials-08-00938-f004:**
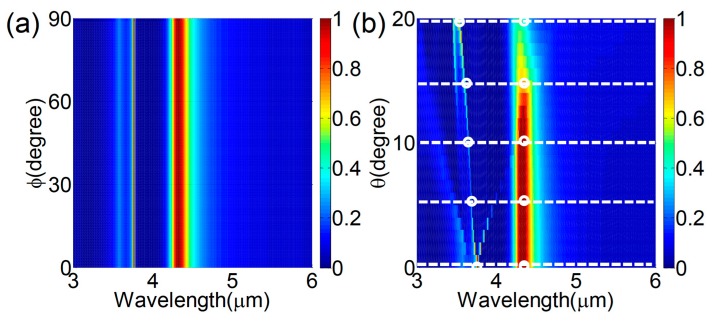
The simulated reflection spectra of the proposed filters with varying (**a**) polarization angles (ϕ) and (**b**) incident angles (*θ*) (ranging from 0° to 20°). The small white circles represent the resonance points of the different incident angles.

**Figure 5 nanomaterials-08-00938-f005:**
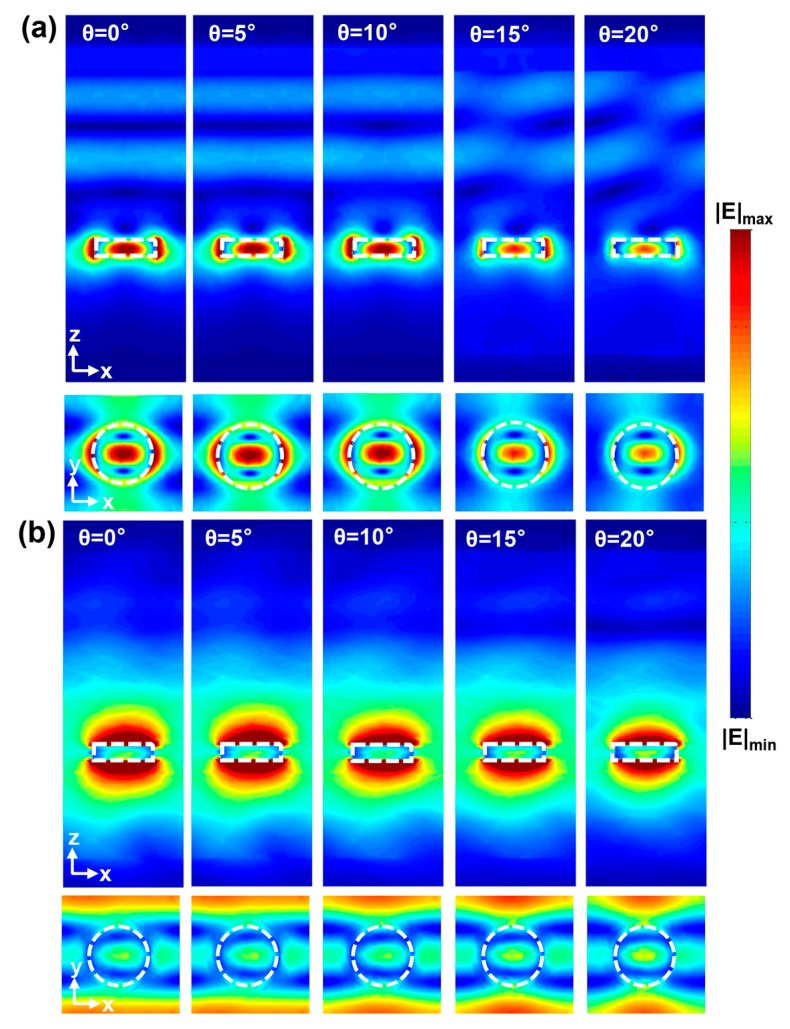
The near-field distributions of the metasurface at the (**a**) main peak (~4.3 μm) and (**b**) side peak (~3.7 μm), with various incident angles.

**Table 1 nanomaterials-08-00938-t001:** Definitions of the parameters used in Equations 1–5.

**p**	Electric dipole moment	**m**	Magnetic dipole moment
$Q_{\alpha \beta }$	Electric quadrupole tensors	$M_{\alpha \beta }$	Magnetic quadrupole tensors
**E**	The electric field	**T**	Toroidal dipole moment
α,β	Cartesian components in the coordinate system	$\varepsilon_{d}$	Relative permittivity of vacuum
$\varepsilon_{r}$	Relative permittivity of Si	c	The speed of light in vacuum

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
