# Peer review of "Dielectric Metasurface-Based High-Efficiency Mid-Infrared Optical Filter"

_nanomaterials, 2018, doi:10.3390/nano8110938_

Reviewer 1 Report

The paper reports on FDTD simulations of a system where Si microdisks are grown in a regular array over a CaF2 substrate. Simulations are aimed in particular at assessing optical behavior in the mid-IR range. The major claim of the paper is that, by changing microdisk diameter, the main peak of reflection spectra, corresponding to almost full reflection, can be tuned in a relatively wide range.

The manuscript is clearly presented and, while topic of dielectric metasurfaces has been widely discussed in recent literature, results may attract some interest owing to he extremely simple architecture proposed. On the other hand, material contained in the manuscript is scarce, at least in purely quantitative terms, resembling more an applicative exercise of a numerical method than a scientific communication.

In my opinion, the work cannot be published in the present form because of substantial lack of original and scientifically sound material. Authors must add other results and improve the presentation in order for the manuscript to be considered for publication. Here in the following a list of suggestions is proposed to this aim.

1.    An array of microdisks is considered, but, as far as I can understand, system behavior is ruled solely by single microstructures. In fact I could not see any result stemming from the regular arrangement of the microdisks such as, for instance, a space resolved reflection pattern simulated in the far field over a transverse scale larger than single microdisk diameter. Such information would be very interesting in view of applications.

2.    As consequence of the comment above, I can’t understand where possible dependence on polarization can occur. In other words, the claimed absence of sensitivity on the in plane direction of the electric field sounds quite expected due to the cylindrical symmetry of the microdisks. Therefore, it is not clear to me why specific attention is deserved by Authors to this point.

3.    As far as applications are concerned, Authors should enter more into the details. Results suggest essentially that the system can be used as good reflector at the design wavelength, but, at the same time, reflectivity is large in a rather large range and is spectrally composed by different narrow and broad peaks [see Fig. 2(b)]. Is there any application, which can eventually benefit from similar reflection spectra (I could not envision any specific area of interest)? And, more in general, how the performance compare with that of standard dielectric mirrors? And how do they compare with other metasurface architectures presented in the literature?

4.    If I well understand, at line 144 and following Authors envision realization of arrays with variable microdisk diameter aimed at achieving strong reflection in a wide range of the mid-IR spectrum. They should simulate the obtained reflected pattern in the far field in order to assess such a proposal.

5.    No detail on the FDTD method is given. It would be interesting to know which algorithms and mesh shape have been used and how constraints have been implemented.

6.    Although paper is limited to simulations, Authors should mention fabrication methods they deem suitable to realize the proposed structure. In addition, they should clarify the advantages of the proposed architecture compared to other, usually more complicated, systems already appeared in the literature.

7.    Authors are encouraged to widen their simulative efforts, for instance by considering variations in the array spacing (I guess it would not affect results unless spacing is considerably smaller by the wavelength, owing to the already mentioned predominance of single microdisk behavior) and in microdisk thickness.  Such results can be relevant to introduce additional degrees of freedom in system design.

8.    Figure 2 plays the most relevant role among the presented results (a large part of the short paper relates to comments to such a figure). First of all, in the pdf version I received figure is presented twice. Furthermore, caption is clearly wrong, legend in panel (c) should use abbreviations consistent with Table I, dashed lines in electric and magnetic field maps must be clarified (possibly, in the caption). However, the most relevant problem with such figure is that I could not understand the relationship between scattering cross section shown in panel (c) and reflectivity in panel (b). Further to a qualitative indication that the broad reflection peak is due to electrical dipole scattering, I can’t retrieve much information from panel (c). Authors must clarify the point and find a physical relationship between simulated scattering cross section and reflection spectra. 

9.    A convincing physical explanation, further to simulation results, should be found and duly discussed also for the different behavior as a function of wavelength for the electrical and magnetic dipole contributions. This is almost completely missing in the present version of the manuscript.    

Author Response

see the attatched file!

Reviewer 2 Report

The authors study numerically the optical properties of high-index dielectric metasurface in the mid-infrared region. They found a regime where such a structure can be used as an efficient optical filter due to high reflectivity at corresponding resonances. They also investigated the oblique incidence and found the limits of incident angles where the reflectivity is still maintained. In general, the results are correct and appear to be interesting to a reader. I just have several comments:

i) In Fig.1 there are four reflected beams for a single incident one. It's confusing and should be removed. Only one reflected beam is enough. If they want to show the diffraction orders (see below), they should be oriented differently.

ii) Fig.2 is shown twice. The narrow reflection resonance near 3.6um is related to diffraction. The authors should investigate this in more details. In Fig. 2.c they show "scattering contributions". I guess this is misleading since these are the amplitudes of corresponding multipoles. Instead, they should reconstruct the reflection spectra from the multipole decomposition to make a clear statement. 

iii) In Fig.3 the authors should indicate the E_max is defined at the resonant condition.

iv) The authors didn't specify which FDTD solver they've used. They should mention it explicitly.

v) Such high-index particles are known to have a strong magnetic dipole resonance. Why the authors don't consider the optical filter based on this type of response?

vi) they should also mention in the text the refractive indices of the used materials.

Author Response

see the attatched file!

Reviewer 3 Report

The paper entitled Dielectric metasurface based high-efficiency mid-infrared optical filter by Fei Shen, Jinghing Wang, Kail Guo, Qingfeng Zhou, Zhongyi Guo discusses the development of a high-efficiency mid-infrared optical filter based on all-dielectric metasurface composed of silicon (Si) nanodisk arrays. By performing numerical simulations, they show that the range of theproposed reflective optical filter could effectively cover a wide range of operation wavelengths by increasing the diameter of the Si nanodisk. The paper is exhaustive with the conclusions supported by data and discussed in the main text. The paper can, however, be improved by improving the English language to correct grammar throughout the manuscript.

Author Response

see the attatched file!

Round  2

Reviewer 1 Report

The Authors demonstrated with their replies to have duly considered most of criticisms raised in my previous review. In several cases, suggestions have been accepted, but related activities deemed out of the scope of the present paper (they will be eventually considered in further works).

Therefore, the manuscript is only slightly improved. 

However, its overall merit is now sufficient and in my opinion it can be accepted for publication in the present form.

Reviewer 2 Report

The authors have taken into account all the comments and suggestions raised by all Rferees and revised the manuscript accordingly.

It can now be published in the present form.